# Drosophila as a Model for Human Viral Neuroinfections

**DOI:** 10.3390/cells11172685

**Published:** 2022-08-29

**Authors:** Ilena Benoit, Domenico Di Curzio, Alberto Civetta, Renée N. Douville

**Affiliations:** 1Department of Biology, University of Winnipeg, 599 Portage Avenue, Winnipeg, MB R3B 2G3, Canada; 2Division of Neurodegenerative Disorders, St. Boniface Hospital Albrechtsen Research Centre, 351 Taché Ave, Winnipeg, MB R2H 2A6, Canada

**Keywords:** *Drosophila* (fruit fly), virus, transposable elements (TEs), neurological, infection, immunity, human, models, transgenic, homologue

## Abstract

The study of human neurological infection faces many technical and ethical challenges. While not as common as mammalian models, the use of *Drosophila* (fruit fly) in the investigation of virus–host dynamics is a powerful research tool. In this review, we focus on the benefits and caveats of using *Drosophila* as a model for neurological infections and neuroimmunity. Through the examination of in vitro, in vivo and transgenic systems, we highlight select examples to illustrate the use of flies for the study of exogenous and endogenous viruses associated with neurological disease. In each case, phenotypes in *Drosophila* are compared to those in human conditions. In addition, we discuss antiviral drug screening in flies and how investigating virus–host interactions may lead to novel antiviral drug targets. Together, we highlight standardized and reproducible readouts of fly behaviour, motor function and neurodegeneration that permit an accurate assessment of neurological outcomes for the study of viral infection in fly models. Adoption of *Drosophila* as a valuable model system for neurological infections has and will continue to guide the discovery of many novel virus–host interactions.

## 1. Introduction

The impact that pathogenic viruses have on human health is constantly evolving and understanding how viral proteins influence cellular homeostasis in vivo provides critical insights into disease pathogenesis and treatment options. Ethical and physiological constraints that limit human models are not barriers in *Drosophila* (fruit fly) systems when examining pathogenicity or for preclinical research purposes. Using the fruit fly as a model organism provides opportunities to uncover specific mechanisms involved in viral pathogenesis and host responses and is a highly manipulable system for genetics and therapeutics. Moreover, *Drosophila* is useful in determining the pathological impact of viral proteins in an intact system.

For over a century, *Drosophila* has been used in research, and the use of this versatile model organism has flourished [1,2]. *Drosophila* has been extensively used to study aspects of human health including cancer, neurodegenerative disease, immune responses, viral infections, and antiviral therapeutics. The ease of maintenance, clear phenotypes, and significant homology with approximately 75% of human disease genes exhibiting comparable sequences in *Drosophila*, make this organism ideal for studying human health and disease [3,4,5,6].

Despite the many advantages *Drosophila* offers as a model for the study of human diseases in general and neurological disorders in particular, some clear disadvantages are the differences in brain anatomy/function and the lack of an adaptive immune system. It is also not straightforward to translate dosage effects used in treatment across the two systems and therefore the fly is an inexpensive and easy-to-manipulate model that can offer preliminary clues that need to be validated in mammalian systems and ultimately clinical trials.

## 2. Antiviral Immunity in *Drosophila*—What to Expect and How It Differs from Humans

Viruses infect every domain of life [7]. Pathogenic human viruses are constantly evolving to evade the immune system, putting pressure on host antiviral mechanisms to evolve alongside. This ongoing arms race provides the opportunity to identify viral neuropathological mechanisms and immune evasion strategies that pertain to neurological disorders. *Drosophila* are a versatile model that can be used to unravel the puzzle of human viral infection and elucidate how antiviral immunity proteins function (recently reviewed in [8,9]). Though humans and *Drosophila* vary somewhat in antiviral immunity activation (see Box 1), they share significant conservation of innate immune signalling pathways and proteins [10,11,12,13] (Table 1). Due to this, *Drosophila* has been used by multiple research groups to identify how different types of human viruses’ function to drive pathology. The *Drosophila* antiviral immune system has been intensively studied and reviewed elsewhere [14,15,16]. This section will provide an overview of the four main branches of the *Drosophila* antiviral immune signalling pathways: Toll, Immune Deficiency (IMD), Jak-STAT and RNA interference (RNAi).

### 2.1. Toll Pathways

First identified in *Drosophila*, the Toll signalling pathway is fundamental in initiating an innate immune response in humans [74,75] (Figure 1). Moreover, Toll signalling is an important aspect of central nervous system (CNS) development and plasticity, by impacting neurite outgrowth, synaptic impulses and promoting either the survival or death of neurons [76]. *Drosophila* have nine Toll and Toll-related proteins [77], whereas Toll signaling in humans is mediated by ten Toll-like receptors (TLRs) [78]. Toll signalling in humans is initiated by pathogen-associated molecular patterns (PAMPs) that directly bind to TLRs (reviewed in [79]). Conversely, activation of classical *Drosophila* Toll signalling is more intricate, as the PAMP is first recognized extracellularly by pattern recognition receptors (PRRs), such as peptidoglycan recognition proteins (PGRP), Gram-negative bacteria binding proteins (GNBP), the serine protease Persephone, and only in select instances through direct binding of the PAMP to Toll proteins. GNBP1, PGRP-SA, and PGRP-SD detect Gram-positive bacteria, whereas GNBP3 detects fungi and Persephone detects fungi-derived virulence factors [12]. Recognition by these PRRs causes the spätzle processing enzyme to cleave spätzle (spz). Cleaved spz binds to the Toll receptor and causes the recruitment of the adaptor proteins dMyD88, Tube, and the kinase Pelle. These proteins form a complex to target and phosphorylate Cactus, the equivalent to the human IκB kinase, causing its degradation. This releases the NF-κB-like transcription factors Dorsal and Dif, allowing their translocation to the nucleus and increases the expression of antimicrobial peptides (AMPs) [12,80] and cytokines [81]. Predictions on which AMPs may act as antiviral peptides have been examined and include Drosomycin, which is induced by Toll signalling in *Drosophila* [82]. It is known that Toll signalling is increased upon viral infection, but individual AMPs are generally not sufficient to defend against viruses alone [80]. Hemocytes are *Drosophila* blood cells and function in innate immunity, injury healing and several forms of the stress response, making them functionally similar to mammalian myeloid cells [83]. AMPs can activate hemocytes, which both strengthens Toll signalling and engulfs infected cells [80]. Another study showed that the Toll signalling pathway in *Drosophila* plays a leading role in the resistance to oral-route-specific viral infections, but not for systemic viral infections [84]. Lastly, the Toll pathway can activate another mechanism of antiviral immunity—autophagy. The autophagy pathway is conserved between flies and humans and can be initiated in *Drosophila* through Toll-7 signalling [85]. This non-canonical Toll signalling cascade is induced by direct viral PAMP interaction with Toll-7 and stimulates an antiviral autophagy response during Rift Valley Fever Virus (RVFV) [85] and Vesicular Stomatitis Virus (VSV) [86] infections in *Drosophila*.

Box 1Key differences in induction of antiviral gene expression between *Drosophila* and humans.The synergy between NF-κB and interferon regulatory factors (IRFs) is key to the induction of innate antiviral immune responses in humans [87,88]. In flies, chronically activated NF-κB signalling leads to increased levels of antimicrobial peptides and is associated with neurodegeneration [89], highlighting the importance of controlled NF-κB activity in homeostasis and infection. A substantial difference in innate immunity in *Drosophila* is the absence of IRF genes and interferon signalling, as IRFs also arose during vertebrate evolution [90,91]. In humans, PAMP recognition by specific TLRs activates an IRF response. IRF3 is a transcription factor particularly crucial in antiviral signalling through initiating the expression of pro-inflammatory cytokines, including the type 1 interferons IFNα and IFNβ [92,93]. These cytokines initiate signalling cascades that induce the expression of hundreds of IFN-stimulated genes (ISGs), many of which have antiviral functions [94,95]. STING signalling in humans also activates both NF-κB and IRF3; however, the activation domain required for these transcription factors is absent in *Drosophila* STING (dSTING), as it arose over the course of vertebrate evolution [96]. Thus, the lack of IRF3 and associated ISGs is a notable difference of innate antiviral immunity in *Drosophila* as compared to mammals and is a limitation for disease modeling in flies.

### 2.2. Immune Deficiency (IMD) Pathway

The IMD pathway is similar to the mammalian tumour necrosis factor receptor (TNFR) pathway (Figure 1). This pathway, like Toll, results in the activation of an NF-κB-like transcription factor, Relish. The canonical IMD pathway is initiated primarily by Gram-negative bacteria at the cell membrane or intracellularly via PGRP-LC and PGRP-LE, respectively [32,75,97]. Once activated, the IMD protein initiates two distinct signalling subdivisions that later converge. In the first IMD division, IMD activates dTAK1, which further activates Kenny (IKKγ) and Ird5 (IKKβ), resulting in the phosphorylation of Relish. The second IMD division requires IMD binding to dFadd, causing the activation of Dredd (homologue of human caspase-8), which is required for the cleavage of the previously phosphorylated Relish into Rel-49 and Rel-68. The N-terminal of Rel-68 is then translocated into the nucleus and regulates the expression of immune genes such as Attacin and Diptericin [75,80,98], which were also identified as antiviral peptides [82].

A non-canonical version of the IMD pathway is also important for immunity against viruses and acts through the *Drosophila* orthologue of the stimulator of interferon genes (STING), dSTING. In mammals, cyclic GMP-AMP synthase (cGAS) is activated by cytosolic dsDNA and produces 2′3′-cGAMP that in turn binds to STING, inducing the nuclear translocation of NF-κB and IRF3 followed by antiviral gene expression [99]. In *Drosophila*, there are two cGAS-like receptors, cGLR1 and cGLR2. The cGLR1 receptor is activated by long dsRNA, whereas cGLR2 is activated by a currently unknown viral stimulus [99,100]. It is known that both cGLR1 and cGLR2 initiate an antiviral immune response by producing cyclic dinucleotides, which occurs similarly in mammals. Likewise, dSTING is triggered by these cyclic dinucleotides, especially 2′3′-cGAMP produced by cGLR2 [100]. As such, 2′3′-cGAMP can protect *Drosophila* against both DNA and RNA viruses and this protection was dependent on dSTING [101]. The function of dSTING is to regulate the expression of antiviral effectors and is positively regulated by dIKKβ and Relish during viral infection. One such effector, Nazo, has been identified by Goto et al. to have antiviral functions against Drosophila C Virus (DCV) and Cricket Paralysis Virus [66], as well as restrict Zika virus infection through dSTING–activated autophagy in the *Drosophila* brain [102].

In addition, the IMD pathways (but not Toll signalling) are essential for recruitment of hemolymph-derived macrophages to the brain in *Drosophila* [103]. Glial cell (but not neuronal) activation of IMD signalling is responsible for drawing macrophages into the inflamed brain tissue. The phagocytic and immune activity of hemolymph-derived macrophages within the brain is further associated with a reduction in locomotor activity and survival.

### 2.3. Jak-STAT Pathway

The first *Drosophila* antiviral pathway discovered was the Janus kinase-signal transducer and activator of transcription (Jak-STAT) (Figure 2). In this pathway, three cytokine-like proteins named Unpaired (Upd) act as ligands for the Domeless (Dome) receptor. Dome shares sequence similarity with mammalian cytokine class 1 receptors, including those in the IL-6 receptor family [45]. Upd1 has roles in development whereas Upd2 and Upd3 are activated by stressors, including viral infection. Upon extracellular Upd2 or Upd3 activation of Dome, two Jak kinases called Hopscotch (Hop; similar to mammalian Jak2 [104]) bound on either side of the cytoplasmic portion of the receptor phosphorylate each other, as well as select tyrosine residues on the cytoplasmic tail of Dome. This creates an attachment site for the SH2 domain of STAT92e (similar to mammalian STAT5 [104]), which is then phosphorylated by Hop. STAT92e is then released from the receptor, dimerizes, and is translocated to the nucleus where it induces the expression of target genes [45,104,105,106]. One of the Jak-STAT-induced target genes is virus-induced *RNA 1* (*vir-1*). This effector gene is strongly induced by DCV infection, but interestingly not bacteria or fungi, indicating this pathway is virus-specific [107]. Null mutant *Hop Drosophila* challenged by DCV have reduced expression of *vir-1,* correlating with a shorter lifespan and a higher viral load [108]. Moreover, this pathway is primarily initiated by viral infection-induced cell damage, not by PAMP recognition [106].

### 2.4. RNA Interference

Arguably the most crucial defense against viral infections in *Drosophila* is RNAi [109] (Figure 2). Consisting of three pathways, RNAi uses microRNAs (miRNAs), small interfering RNAs (siRNA), and Piwi RNAs (piRNAs) to defend against viruses of varying origins. The first two RNAi pathways defend against dsRNA viruses and utilize the *Drosophila* Dicer (Dcr) and Argonaute (AGO) proteins to silence viral gene expression. The miRNA pathway uses the RNase III enzyme Drosha, and the dsRNA binding protein Pasha to process the primary miRNA transcripts into precursor miRNAs (pre-miRNAs) in the nucleus. The pre-miRNA transcripts are exported to the cytoplasm where they are recognized and cleaved by Dcr-1 (RNase III enzyme) with assistance from R3D1 (a dsRNA binding protein). This process creates 22–23 nucleotide long miRNAs, which are loaded onto AGO1, forming the miRNA-induced silencing complex (miRISC). One of the miRNA strands is discarded, and the other is incorporated into the miRISC and acts as a guide to target and bind partially complementary sequences of viral mRNA, thus inhibiting its translation and silencing viral gene expression [14,106,110,111,112]. The siRNA pathway shares a similar mechanism to that of the miRNA. The siRNA pathway is initiated by the recognition and cleavage of viral dsRNA into 21-23nt siRNA duplexes by Dcr-2 and the dsRNA binding protein R2D2. These siRNA duplexes are loaded onto AGO2 by the Dcr-2/R2D2 complex [112]. The passenger strand is removed, and the guide siRNA strand is stabilized by 2′-*O*-methylation by dHen1, forming the mature siRISC. The siRISC identifies fully complementary mRNA to be cleaved by the AGO2, thus preventing viral protein synthesis [110,111,112,113].

There are not only external viral threats; many viral signatures stem within host DNA. Transposable elements are mobile genomic remnants of previous viral infections that remain in the DNA of its host. PiRNAs are non-coding RNAs generated from discrete genetic loci containing defunct transposon sequences and are used in both *Drosophila* and humans to silence transposons and maintain the integrity of the genome [114,115,116]. There are over 8 million piRNAs found in humans and nearly 42 million found in *Drosophila* [117]. Unlike miRNAs and siRNAs, piRNAs are generated by ssRNA precursors [112], are longer at 24–32 nucleotides, are produced independently of Dicer, and utilize AGO proteins from the PIWI clade [117]. In this pathway, piRNA are transcribed from a piRNA cluster and loaded onto one of the *Drosophila* PIWI proteins: Piwi, Aubergine (Aub) or AGO3. Piwi and Aub preferentially bind piRNAs containing the antisense strand of transposons, whereas AGO3 incorporates piRNAs with the sense strand orientation [114]. The mature nuclear Piwi–piRNA complex can directly degrade transposon mRNA. The cytoplasmic Aub–piRNA can initiate the piRNA ping-pong cycle to amplify the piRNA signal. Aub–piRNA cleave the sense strand of transcribed transposon RNA, not only silencing it but also forming secondary piRNAs that are loaded onto AGO3 and promote the generation of a new cluster of piRNAs [117].

This pathway is also of interest due to the utilization of the transcription and export (TREX) complex in both *Drosophila* and humans. TREX has similar functions in both species, including the biogenesis of piRNA, preventing R-loop formation and insuring genome stability [72,118]. Mutations in *Drosophila* TREX causes defective piRNA biogenesis, leading to derepression of transposons and as a result faulty gametogenesis and male sterility [72]. On the other hand, mutations in human TREX can lead to cancers, neurodegenerative disorders, and autoimmune and inflammatory diseases [118,119]. TREX mutation or disruption in disease states may be linked to the derepression of endogenous retroviruses (ERVs). In *Drosophila,* ERVs such as Gypsy are derepressed by defects in the piRNA pathway [120] and can infect neighbouring cells [121]. The same occurs in humans, where ERVs make up approximately 8% of the human genome. ERVK is the most recent retroviral acquisition to the human genome, and control over its reverse-transcribed DNA is mediated by TREX1, with mutations in the human TREX gene leading to autoimmune disease, including the inherited encephalopathy Aicardi–Goutières syndrome and chilblain lupus [73].

## 3. Neuroinfection and Neuroimmunity Models in *Drosophila*

There are many different strategies to study the impact of viral infection in *Drosophila*. While using viruses that naturally target *Drosophila* represents a clear pathogen–host interaction, flies can also be infected with zoonotic pathogens, including human viruses [122]. Using a more reductionist approach, transgenic *Drosophila* expressing individual viral genes or proteins can be used to assess the relationship between viral products with the host in terms of both molecular pathology and behavioural effects. In many instances, models exhibiting robust virus-driven phenotypes can be used to test the efficacy of antiviral drugs. Below, we will highlight some of the many viral models in *Drosophila*, which illustrate how flies can provide insight into human neurological disease [123].

### 3.1. In Vitro Drosophila Viral Neuroinfection Models

The use of *Drosophila* cell lines for in vitro culture is a common approach to study viral infection. As recently reviewed by Luhur et al. [124], there are many widely available *Drosophila* cell lines and in conjunction with somatic cell genetics, there is much possibility to generate novel lines (Figure 3). For the study of immunity at the host–pathogen interface, hemocyte-derived lines are frequently used, as their innate immune responses are similar to that of vertebrate macrophages, and they are essential for control of virus infection and brain immunity. Schneider lines are also prevalent, and they are often used to probe for viral receptor–ligand interactions and modulation of signalling cascades. Below, we will highlight examples of the benefits and drawbacks of using *Drosophila* cell lines to study viral neuroinfections.

#### 3.1.1. Studying Viral Entry

Severe cases of Herpes Simplex Virus 1 (HSV-1) infection can lead to meningitis and encephalitis [125]. The use of *Drosophila* cells for studying infection is not always straightforward, as seen when Fan et al. investigated the requirements for HSV membrane fusion in Schneider 2 (S2) cells. Unfortunately, it was found that none of the tested S2 cell receptors were able to facilitate HSV-1 membrane fusion. This indicates that S2 cells are missing the required receptor to mediate cell fusion [126]. The inability for HSV-1 to enter S2 cells does not mean that *Drosophila* cells are not useful to study this virus. Liu et al. developed a protocol to generate the expression of human surface ligands and receptors on S2 cells. Molecules expressed using this high yield S2 sub-clone method included the Human Herpes Virus Entry Mediator (HEMV), CD160, Fas, LIGHT, DcR3 and B7H3. HEMV is of particular importance as a receptor for the HSV-1 glycoprotein D (gD) and mediating viral cell entry. By enhancing the production of HEMV 20-fold in S2 cells, this protocol allowed researchers to produce increased amounts of human protein for downstream purification, as well as perform viral ligand–receptor interaction studies at a lower cost [127]. S2 cells have also been modified to produce recombinant HSV-1 gD1. Results of experiments performed by Mao et al. showed that the gD1 ectodomain is secreted while the full-length protein is restricted to the lipid membrane. Mice immunized with the purified gD1 ectodomain generated high levels of antibodies, proving that S2 cells can synthesize immunogenic viral proteins [128].

Sindbis virus (SINV) infects neurons and causes viral encephalomyelitis (inflammation of the brain and spinal cord)*,* both in humans and in rodent models [129,130]. *Drosophila* were used to uncover SINV entry mechanisms and treatment options. Rose et al. discovered that the Natural Resistance-Associated Macrophage Protein (NRAMP), a transmembrane metal ion transporter, was required for SINV binding and entry into *Drosophila* cells, but not VSV or WNV, indicating this mechanism is SINV specific [131]. Null mutant dNRAMP flies displayed a 4-log reduction in SINV viral titers. Treatment of *Drosophila* cells with iron resulted in the attenuation of SINV infection as NRAMP is post-translationally degraded in the presence of high iron, resulting in a reduction of cellular receptors for SINV. The vertebrate dNRAMP human homologue, NRAMP2, also mediated SINV binding and entry into mammalian cells. This work highlights how a loss-of-function approach in flies can assist in the identification of unknown viral receptors.

Similarly, a screen of the *Drosophila* kinome using S2 cells identified AMP-Activated Kinase (AMPK) as a key factor facilitating Vaccina Virus (VACV) entry into cells through modifying the actin cytoskeleton and facilitating micropinocytosis uptake [132]. In addition, the VACV multiprotein entry-fusion complex and a low-pH environment are also necessary for VACV entry to S2 cells [133]. However, this is an example of where the use of *Drosophila* cell lines is limited to examining viral entry, as VACV is unable to replicate in insect cells. Therefore, a caveat (and depending on experimental design, a benefit as well) of employing fly cell lines for modeling human neurological infections is the differential cellular composition, which does not always support viral entry or viral replication.

#### 3.1.2. Assessing Viral Replication

Although Hepatitis B Virus (HBV) is often known for its association with viral hepatitis, it can also cause peripheral neuropathy in approximately 5% of cases [134]. *Drosophila* S2 cells have been used to uncover HBV replicative mechanisms. The function of the viral nucleocapsid minimal promoter element was modeled using transient transfection of S2 cells. It was found that the HBV nucleocapsid promoter has three Sp1 binding sites that contribute to the degree of viral transcription [135]. Individual Sp1 sites within the HBV nucleocapsid promoter have varying effects on viral transcription, which was further confirmed in human Huh7 hepatoma cells [136,137]. Binding to the first Sp1 site did not affect HBV transcription, the second Sp1 site negatively regulates viral transcription, and engagement of the third Sp1 site within the ENII enhancer upregulates the expression of all HBV genes [136]. While this demonstrates reliance on cellular transcription factors for HBV transcription, HBV and host gene expression can also be impacted by the HBV X protein viral transactivator. S2 cells stably transformed with the HBV X gene showed that the HBV X protein can activate all three classes of RNA polymerase III promoters by mediating the increase of the TATA-binding protein (TBP), which is a limiting transcription factor in the cell [138]. Using S2 cells, Wang et al. subsequently investigated whether the HBV X protein functions were reliant on the Ras/Raf1 signalling pathway. Results showed that no HBV X protein driven increase in TBP occurred in Ras null mutant S2 cells, highlighting the necessity of the Ras pathway for HBV X activation [139]. Again, these results were replicated in human Huh7 cells [140], demonstrating that *Drosophila* cell lines can generate comparable results to mammalian cell culture in select contexts of host–pathogen interaction.

#### 3.1.3. RNAi Screens to Identify Antiviral Pathways

Rift Valley Fever Virus (RVFV) is a zoonotic arboviral pathogen that can cause brain infections, resulting in severe headaches, visual disturbance and confusion, and in severe cases meningoencephalitis (inflammation of the brain and tissues lining the brain) [141]. Using a genome-wide RNAi screen in fly DL1 cells, Hopkins et al. identified the mRNA decapping enzyme Dcp2 as a RVFV restriction factor [63]. As part of their replication cycle, Bunyaviruses steal the 5′ ends of host mRNAs (“cap-snatching”) to facilitate 5ʹ capping of their viral mRNAs. Dcp2 restricts RVFV infection in *Drosophila* by decapping mRNAs preferentially targeted by RVFV cap-snatching, such as those related to cell cycle. Without competition against capping viral mRNAs, Dcp2-depleted flies succumbed to RVFV infection, and these flies also showed increased viral replication. Interestingly, it was found that Dcp2 restricts RVFV specifically, as removal of Dcp2 had no effect on infection with other RNA viruses (DCV, SINV, VSV) from disparate families.

FoxK1 is a transcription factor in humans that is important for the development of cortical structures [142]. Panda et al. performed a *Drosophila* RNAi genetic screen that identified the FoxK (fly orthologue of FoxK1) transcription factor as having antiviral activities against neurotrophic SINV [65]. The action of FoxK was found to coordinate with the antiviral protein Nup98. Mechanistically, Nup98 binds to promoters of several virus-induced genes to transcriptionally elicit an antiviral response [68]. FoxK knock-out flies showed increased SINV replication and FoxK-depleted *Drosophila* DL1 cells also showed increased SINV, VSV and RVFV replication [65]. A decrease in FoxK also resulted in reduced expression of Nup98-dependent genes, confirming that FoxK acts in concert with Nup98 to regulate the expression of antiviral genes. FoxK is highly conserved between flies and humans [65]. Follow-up human cell culture experiments showed increased SINV expression in FoxK1 knockdown cells as compared with U2OS cells transfected with non-targeting siRNA, confirming a role for FoxK in regulating cerebral antiviral gene expression and viral replication in both *Drosophila* and humans [65].

An RNAi screen was also used to find genes involved in influenza virus replication [143]. Fly D-Mel2 cells (embryo-derived) do not allow for influenza virus entry. To overcome this, the virus was genetically modified by replacing the *hemagglutinin* and *neuraminidase* genes with the VSV glycoprotein G, which allows virus entry into both mammalian and fly cells. The modified influenza virus could then infect D-Mel2 cells and was used to identify over 100 candidate genes important in influenza replication, through an RNAi library screen of over 13,000 genes (90% of the *Drosophila* genome). Of interest were fly genes with human homologues known to impact influenza replication, including the mitochondrial export complexes III and V and the genes *ATP6V0D1* (encodes subunit D of the V-ATPase proton pump)*, NFX1* (nuclear export factor) and *COX681* (electron transport). This approach further reveals the feasibility of using genome-wide RNAi screens in *Drosophila* to identify genes involved in viral replication. Moreover, the identification of new viral restriction factors through screening is an effective strategy for identifying novel antiviral drug targets [144].

#### 3.1.4. Examining Antiviral Immune Responses In Vitro

RNAi is a well-established antiviral mechanism usually associated with immunologic response to dsRNA. Next-generation sequencing analysis of *Drosophila* S2 cells infected with RVFV display virus-derived small interfering RNAs (vsiRNAs) [145]. VsiRNAs are processed and function similarly to siRNAs and are formed from viral RNA during active infections [146]. The identified 21–22 nucleotide vsiRNAs corresponded to all three RVFV gene segments. Being a key component of the RNA-induced silencing complex (RISC) pathway, Ago2 knockdown flies displayed increased viral titers and reduced lifespan, showing that the antiviral pathway responsible for the response to RVFV is RNAi [145].

Vesicular stomatitis virus (VSV) is another cause of acute infection of the central nervous system in rodents and non-human primates [147]. Mueller et al., showed that RNAi was initiated by VSV in *Drosophila* Kc167 cells (derived from embryos) even though VSV did not produce a detectable amount of dsRNA in fly cells [148]. This then posed the question of how RNAi was initiated if not through dsRNA recognition. Flies mutant for defective RNAi genes (*Dcr-2, R2D2, AGO2*) had increased VSV titers that lead to death within 12 days. VSV-derived vsiRNA was detected in infected S2 cells that control VSV RNA amounts [148].

#### 3.1.5. Production of Viral Proteins for Therapeutics Using Drosophila Cells

*Drosophila* cells have also been successfully used to develop vaccine components for infections leading to neurological complications, such as rabies virus [149], HIV [150] and HBV [134]. Multiple research groups have used S2 cells to produce the recombinant rabies glycoprotein for use in developing vaccines and in diagnostics [151,152,153]. Similarly, S2 cells have been useful in purifying and producing vaccine components including production of the HBV small surface antigen [154]. The S2 cell-produced viral antigen was over 98% pure and successfully initiated both an antibody and cytotoxic T lymphocyte response in immunized mice [155]. The HBV X protein has also been used to develop a therapeutic vaccine that increases viral clearance in a mouse model [156]. Similar work has been performed in HIV models, with the development of S2 cells that can secrete a truncated form of HIV gp120 [157]. The S2 cell-produced gp120 was glycosylated, could bind the CD4 receptor, was detected by a gp120 antibody and could prevent the formation of syncytia between HIV infected cells and uninfected CD4 cells. The recombinant fly synthesized protein accurately resembles the full HIV gp120 protein and can therefore be used for further scientific analysis. As an extension of this strategy, Yang et al. generated an HIV envelope, Gag and Rev protein containing plasmid to produce transfected S2 cells that secreted HIV virus-like particles [158]. The virus-like particles produced spike proteins that were recognized by neutralizing antibodies, and mice primed with DNA and inoculated with VLPs initiated an antibody and CD8^+^ T cell immune response. Taken together, it is clear that *Drosophila* cells can not only be used to investigate host–pathogen interactions, but they are also a resource for therapeutic and diagnostic development.

### 3.2. In Vivo Drosophila Viral Neuroinfection Models

Viral infection of *Drosophila* in vivo can be performed using fly-specific pathogens, as well as select human pathogens. As recently reviewed by Harnish et al. [8], *Drosophila* are well-suited to study host–pathogen interactions and perform large genetic and pharmacologic screens. Indeed, viral pathogenesis and endpoints have been well documented in flies; we refer the reader to a review by Xu and Cherry for a comprehensive compilation of viruses studied in *Drosophila* [16]. However, most viral infection models in flies are not focused on neurological outcomes. Nonetheless, novel approaches in genetic manipulation [159], behavioural assessment (automated monitoring systems [160]) and neuropathology (FlyClear tissue clearing protocols [161]) are expanding experimental design beyond the conventional readouts used to profile neuroinfection and immunity outcomes in flies (Figure 1). Below, we will highlight examples of the benefits and drawbacks of using virus-infected *Drosophila* to study related neurological sequelae.

#### 3.2.1. Examining Antiviral Immune Responses In Vivo

Epstein Barr Virus (EBV) is associated with infectious mononucleosis [162] and multiple cancers [163,164,165], although recently its link to increasing the risk of developing Multiple Sclerosis highlights EBV’s importance for neuroinfection [166,167]. A study by Sherri et al. examined pro-inflammatory pathways induced by EBV DNA [168]. Wild-type *Drosophila* injected with EBV DNA showed a 115-fold increase in the AMP diptericin produced by the IMD pathway [168]. The IMD pathway also plays a part in the cellular response against EBV, shown by increased hemocyte numbers in response to EBV DNA [168]. Similar work in a gastrointestinal model of EBV DNA-driven inflammation in flies also showed enhancement of inflammatory response by intensifying IMD signalling [169]. In humans, members of the TNFR superfamily (similar to *Drosophila* IMD) also function in the immune response against EBV-infected cells [170,171], showcasing the similarities between EBV-induced immune signalling in *Drosophila* and humans.

Liu et al. investigated the role of Zika virus (ZIKV) on brain inflammation using *Drosophila* [172]. There was increased ZIKV in the fly brain over time, indicating that this virus can successfully infect and replicate in flies. It was also found that *diptericin*, a target gene of the IMD pathway, was increased upon ZIKV infection but not *drosomycin*, a Toll pathway target. Indeed, Relish (NF-κB homologue; induced by IMD signalling) null mutant flies had increased ZIKV titers, increased lethality, and decreased activation of *diptericin* in neurons or glia [172]. The Stimulator of Interferon Genes (STING/dSTING in flies) has conserved antimicrobial roles in flies and humans [42], and its activation is Relish-dependent during ZIKV infection [172]. As discussed next, STING is a key hub not only for antiviral signalling cascades, but also directly ties into viral clearance processes by acting as a scaffold for LC3 lipidation during autophagy [173].

#### 3.2.2. Investigating Viral Clearance In Vivo

Autophagy is an intrinsic innate immune mechanism that degrades cytoplasmic cargo and plays a role in protecting host cells from pathogens [174]. In many instances, autophagy is protective against viruses, but not in all cases [175]. Deregulation of the autophagic response is also a common issue in neurodegenerative disease [176]. STING is protective against ZIKV by inducing autophagy [102]. To demonstrate the connection between immune signalling and autophagy, *Relish* and *dSTING* mutant flies were infected with ZIKV and evaluated for Atg8-II accumulation, which is indicative of an autophagic response. With loss of either Relish or dSTING expression, there was no evidence of increased Atg8-II during ZIKV infection of flies, indicating that both Relish and dSTING are required for ZIKV-induced autophagy. Liu et al. further show that depletion of Ref(2)P (orthologue of human p62; autophagy cargo receptor [71] and known viral restriction factor in flies [177]) results in enhanced ZIKV load in flies [172].

The GAL4/UAS system is commonly used in *Drosophila* to conduct in vivo screens of gene expression manipulation. This is achieved by crossing transgenic flies that express the yeast GAL4 protein under the control of tissue or cell-specific promoters with a respondent transgenic that carries the yeast UAS sequence, to which GAL4 binds, followed by a gene to be turned on or knocked down (i.e., RNAi). To determine whether ZIKV clearance was cell-specifically mediated, Atg5 was depleted in neurons using *elav-gal4* and glia using *repo-gal4*. Loss of autophagy in either cell type was associated with increased ZIKV replication. Finally, when fed rapamycin (an autophagy inducer) prior to ZIKV infection, flies were protected after challenge with ZIKV [102,172]. Through drug screening approaches, human neuronal models of ZIKV infection corroborate the importance of autophagic mechanisms in the control of neuroinfection [178,179]. While humans do not share a high degree of upstream NF-κB-activating signalling components with flies, this example demonstrates that downstream autophagy induction through the STING pathway is an ancestral and conserved mechanism involved in antiviral response.

In contrast, while dSTING was restrictive against Drosophila C Virus (DCV) infection in flies, Relish and Atg mutants were no more prone to DVC infection than control flies [172]. Thus, despite the reliance on dSTING to control DCV infection, this effect was not mediated through NF-κB signalling or autophagy. This speaks to virus-specific interaction with host machinery and the possibility that dSTING confers antiviral resistance through additional mechanisms. This further highlights dSTING as having a major role in viral clearance.

Additionally, Shelly et al. showed that in flies, autophagy had an antiviral effect against VSV [180]. It was demonstrated that the PAMP that initiated an autophagic response during infection was the surface glycoprotein VSV-G. Autophagy regulated VSV replication in both adult flies and S2 cells. Furthermore, flies with loss of Atg18 (required for autophagy) function were more susceptible to VSV infection, but not DCV. Ongoing autophagy effectively reduced viral replication, and conversely a deficiency in autophagy led to increased viral replication and pathogenesis. These results suggest that intrinsic autophagy deficits in neurodegenerative disease may not only contribute to a failure to control cellular proteinopathy, but pathogens as well.

#### 3.2.3. Investigating Viral Latency In Vivo

Fly models have also been useful to assess the epigenetic control over viral latency. HSV-1 latency-associated transcript (LAT) is the only gene transcribed during HSV-1 latency. Chen et al. showed that an intron within LAT contains an 800-bp region (consisting of 16-bp repeats) that acts as an insulator preventing inactive chromatin from silencing the viral LAT promoter and thus acting as a chromatin boundary to separate active and repressive chromatin [181]. When LAT insulators and embryonic enhancers were injected into *Drosophila* embryos, it was found that the insulator protected the LAT transgene from positional effects, caused by transgene integration in the eye. Various eye colours were observed in adult *Drosophila* carrying the control transgene, whereas *Drosophila* with the LAT transgene had little variation in eye colour in the presence of enhancers [181]. Mechanistically, *Drosophila* dCTCF (orthologue of vertebrate CTCF transcriptional regulator) associated with the CTCCC sequence in the LAT repeats to regulate chromatin activity. The deletion of the repeats impaired LAT insulator activity in embryos, and knockdown of dCTCF interfered with enhancer blocking activity of the LAT insulator in transfected Schneider 3 (S3) cells [181].

#### 3.2.4. Infectious Neurodevelopmental Models

Viral infections can trigger neurodevelopmental disturbances [182]; one that has recently received much attention has been those reported following Zika virus infection. ZIKV infection in pregnant women can cause microcephaly (a birth defect resulting in smaller than typical head size and brain atrophy) in newborns [183]. *Drosophila* have been a useful model to assess why amongst Flaviviruses, ZIKV drives microcephaly pathology. Comparing flavivirus–human protein-interaction networks, Shah et al. identified a putative association between ZIKV NS4A and cellular protein Ankyrin Repeat and LEM Domain Containing 2 (ANKLE2) [184]. ANKLE2 mutation is known to cause autosomal recessive microcephaly in humans [185]. Using transgenic UAS-NS4A flies crossed with neuronal-Gal4 drivers, 3rd instar larval progeny exhibited a significant decrease in brain volume compared to controls [184]. Co-expression of ZIKV NS4A with hANKLE2 or dAnkle2 rescued this phenotype, but not with the microcephaly-associated hANKLE2 variant (hANKLE2 Q782X), indicating that ANKLE2 is protective against ZIKV-driven neuropathology. In contrast, Dengue virus serotype 2 (DENV2) NS4A induces a milder reduction in brain volume that was not statistically different from wild-type brains. NS4A expression in the background of heterozygous Ankle2^A^/+ mutant flies resulted in a more severe phenotype, with reduced numbers of neuroblasts and a clear impact on brain volume and development. A complementary study by Link et al. also demonstrates that loss of ANKLE2 function results in disruption of the Par complex, which is required for proper asymmetric division of neuroblasts in *Drosophila* [186]. Thus, loss of ANKLE2 as a result of microcephaly-associated genes or congenital infection may contribute to microcephaly by impacting neural cell proliferation or the fate of neurons during development. In human fetal stem cells, ZIKV NS4A and NS4B proteins (but not corresponding DENV proteins) inhibit neurogenesis and promote an autophagic response through cooperative viral protein disruption of the Akt-mTOR pathway [187], further strengthening the above findings in flies.

#### 3.2.5. Infectious Neurodegenerative Models

Human Immunodeficiency virus (HIV) can cause a spectrum of neurocognitive deficits labeled under the umbrella term HIV-associated neurocognitive disorder (HAND) [150,188]. HIV can enter the CNS during early infection, and the brain can be a site of HIV replication [188]. Prolonged HIV infection and inflammation in the CNS contributes to the development of HAND. Neurodegeneration is often associated with heightened neuronal aneuploidy (loss and/or gain of chromosomes) [189,190]. Aneuploidy is postulated to arise from mis-segregation of DNA in neuronal progenitor cells, although there is also evidence for cell cycle re-entry as a driver of aneuploidy in neurodegenerative disease [190]. Battaglia et al. aimed to determine the outcome of the HIV Tat-tubulin interaction through HIV Tat injection into fly syncytial embryos [191]. The result was mitotic delays from HIV Tat’s impact on kinetochore alignment and the timing of sister chromatid separation. Moreover, HIV Tat expression in larval brains led to increased polyploid and aneuploid cells. This work suggests that mitotic spindle checkpoints can be overridden by the impact of HIV Tat on tubulin during cell cycling. A later publication showed that the HIV Tat interacts with the ribosomal protein S3 (RPS3), which is involved in modulating microtubule dynamics and α-tubulin assembly [192]. HIV Tat was found to bind to and increase RPS3 nuclear expression, which led to impaired assembly of α-tubulin. RPS3 also co-localized with α-tubulin around the chromosomes at the mitotic spindle. This further confirms that HIV Tat interferes with the mitotic spindle and chromosome assembly in mitosis, which may have implications for the progression of HAND.

Several human picornaviruses are associated with neurological motor disturbances, including poliovirus and coxsackievirus B3 [193]. A picorna-like virus called Nora virus has recently been shown to cause locomotor disturbance in *Drosophila* [194]. While infection did not impact lifespan, climbing (geotaxis) assays in Nora virus (and DCV) infected flies were significantly impaired compared with uninfected controls. It is postulated that Nora virus may impact fly brain function as a result of Nora virus trafficking from the primary site of infection in the gut through circulation in infected hemolymph [195]. Nora virus-infected *D. melanogaster* also show enhanced immune-related gene expression over time, with enhancement of Toll, IMD and hematopoiesis genes, in conjunction with decreased RNAi-related genes [196]. It remains unclear whether perturbed immune response may be tied to motor dysfunction in Nora virus-infected flies.

### 3.3. Viral Transgenic Drosophila

As described by Hughes et al. [197], the GAL4/UAS system is a widely used approach to create transgenic *Drosophila* expressing viral genes. Using this strategy, the viral gene of interest is placed under the control of an upstream activating sequence (UAS), which is activated by binding of the GAL4 transcription factor. Crossing UAS viral transgene responder flies with a select GAL4 driver will result in transgene-expressing and littermate control progeny. The tissue specificity of the viral transgene can be further controlled based on the type of GAL4 driver selected, such that pan-neuronal (ELAV-Gal4), motor-neuron-specific (D42-Gal4) or glial (Repo or Alarm-Gal4) expression can be achieved. However, as described below, other strategies exist to drive (or eliminate) viral protein expression in neural tissues.

While most viral transgenic models seek to insert novel viral protein genes into the fly genome to study viral pathology, CRISPR technology also allows for selective addition or removal of genes, even in existing GAL4/UAS models [159]. This approach has been used in flies to selectively knock-out genes in neural tissues and perform rescue experiments, through UAS-controlled Cas9 expression in conjunction with specific guide-RNAs [159]. CRISPR technology has also been used to ablate the expression of viral transposable elements in mammalian genomes [198], although this approach has not yet been used in flies.

More conventionally, Adamson et al. generated transgenic fly models to study the interaction between the EBV immediate early genes BRLF1 (R) and BZLF1 (Z) and host genes [199,200]. These viral genes were cloned into a P-element vector (pGMR; Glass-mediated response) allowing for eye-specific expression of R or Z. R and Z are viral transcription factors that function in initiating transcription of EBV early genes in the lytic phase of viral replication [200]. R and Z activities in the developing eye displayed identifiable but opposite phenotypes in *Drosophila*. R expression induced a 1.5-fold increase in the number of cells entering cell division, leading to over-proliferation of eye tissue and a rough eye phenotype. The ommatidia (surface of the compound eye in fly) also became unorganized in heterozygotes or diminished in homozygote flies. Conversely, Z expression caused a loss of ommatidia, leading to a smooth eye phenotype and a full loss of eye pigment. Thus, the Z protein functions to inhibit the cell cycle, increase apoptosis, and prevent cone and pigment cell differentiation in the *Drosophila* eye [199], whereas R leads to further cell proliferation. Interestingly, when R and Z were co-expressed, Z was able to reverse the over-proliferation in the eye [200]. This example highlights how alternative systems to GAL4/UAS can be effectively used in viral *Drosophila* models.

Nonetheless, the GAL4/UAS system has been an asset to the study of viruses such as HIV. Lee et al. used HIV Nef transgenic flies to determine the signalling molecules responsible for the development of HIV pathogenesis [201]. Nef expression reduced wing and larval-wing imaginal disc size. Through genetic interaction analysis, Nef was found to interact with basket (*Drosophila* equivalent of JNK). Expression of *bsk* (encoding basket) with Nef exacerbated the wing phenotype compared to Nef alone. It was found that Nef induced caspase-dependent apoptosis via JNK pathway in transgenic fly wings. Nef was also shown to inhibit Relish innate immune signalling in a JNK-independent manner. These results in *Drosophila* are in stark contrast to Nef-over-expression experiments performed in human promonocytic cell line U937, whereby exogenous Nef activated NF-κB [202]. Additional human and murine studies point to HIV Nef as an activator of NF-κB [203,204]. Together, these results caution researchers to employ parallel models and validate fly models with complementary human systems.

A greater consistency across systems can be seen when modelling the cellular impact of HIV Vpu. Leulier et al. used Vpu transgenic flies to demonstrate that this viral protein inhibits Toll pathway activation by stopping the degradation of Cactus (homologue of IκBα; inhibitor of NF-κB) [205]. Mechanistically, the prevention of Cactus degradation likely hinges on the ability of Vpu’s phosphorylated DSGXXS motif to mimic the destruction motif that is also found in both mammalian and fly IκBα proteins [205,206]. Vpu interacts with the human β-TrCP, and it was found that this interaction was conserved between Vpu and the *Drosophila* β-TrCP orthologue SLIMB [207]. This allows HIV Vpu to interact with the E3 ubiquitin ligase β-TrCP, thus interfering with the IκBα/β-TrCP interaction, which is necessary for Cactus/IκBα degradation. Additional studies support HIV Vpu’s role in the stabilization of IκBα as an important factor for limiting the nuclear translocation of NF-κB p65 and quenching innate antiviral response in human models [208,209].

### 3.4. Antiviral Drug Screening in Drosophila

One of the influenza virus proteins necessary for cell entry is the matrix protein 2 (M2) [210]. The LaJeunesse group developed transgenic *Drosophila* for the influenza A virus *M2* gene, which functions as a proton channel by altering intracellular pH [211]. Via the UAS/Gal4 system, expression of M2 produced a rough eye phenotype and wing-vein defects (loss of anterior cross vein); visualization of phenotypes is presented in the paper by Adamson et al. [211]. The cellular location (plasma membrane and intracellular compartment membranes) and function of M2 appeared to be the same in flies and mammals, making this model appropriate for influenza drug screening. To rescue M2-induced phenotypes, the anti-Influenza drug amantadine was administered to larvae. Amantadine treatment caused a reduction in M2-induced rough eye phenotype in adult flies in a dose-dependent manner. A genetic screen using M2 transgenic flies found that deficiencies in genes encoding subunits of the vacuolar V1V0 ATPase worsened the M2 rough eye phenotype. In complement, follow-up mammalian cell culture experiments inhibiting the V1V0 ATPase in MDCK cells resulted in a decrease in the number of influenza-infected cells. Interestingly, the M2 protein can be further modified to a toxic M2 (H37A) form that has lost its proton selectivity, which induces cell death. Lam et al. used this toxic M2 protein to selectively remove *Drosophila* imaginal discs, hemocytes, heart cells and nervous tissue for the study of fly development [212].

Neurological symptoms have been frequently reported in patients infected with severe acute respiratory syndrome-coronavirus-2 (SARS-CoV-2) [193,213], the causative agent of COVID-19. Presentations can range from a loss of taste and smell, seizures, stroke, epilepsy, meningitis/encephalitis and motor disturbances. COVID-19 mammalian research models have a host of limitations including high maintenance costs, few progeny, long reproduction times and limited viral protein and drug screening capabilities. In a situation where time is of the essence, *Drosophila* models may be a faster and more efficient option [214]. In fact, bioinformatic analysis showed SARS-CoV-2 virus–host interactome is 90% conserved between humans and fruit flies [215]. Based on this information, transgenic *Drosophila* were generated to screen COVID-19 genes of interest. Using the Gal4-UAS system, expression of the SARS-CoV-2 genes Orf6, Orf7 and Nsp6 all individually caused disease phenotypes, exhibited by developmental lethality, reduced viability, abnormal wing positions, locomotor defects and reduced tracheal branching [215]. As it was previously known that Orf6 interacts with the NUP98–RAE1 nuclear pore complex [67,216], flies were fed Selinexor, an FDA-approved selective inhibitor of nuclear export protein XPO1 [217]. Similar to Selinexor, reduction of Orf6 cytotoxic effects in a human HEK 293 T cell in vitro culture [216], flies administered Selinexor displayed attenuation of all Orf6-induced disease phenotypes [215]. Clearly, *Drosophila* have been a valuable resource to study SARS-CoV-2 pathology and drug efficacy, as mice are not naturally susceptible to this virus and larger mammals have limited drug testing capabilities (due to ethical constraints) [214].

*Drosophila* have also been used to screen for negative symptoms associated with antiviral therapy. While highly effective at controlling HIV infection, anti-retroviral therapy (ART) is also associated with the development of peripheral sensory neuropathy (PSN) in select patients [150]. Bush et al. demonstrated how ART with AZT (Zidovudine or Azidothymidine) can induce thermal and mechanosensory nociceptive hypersensitivity in larvae [218]. This sensitization was associated with degeneration in sensory neurons, as measured by an increase in fragmented terminal (distal) dendrites in C4da neurons. These findings are consistent with decreased nerve fiber density in patients with HIV neuropathy [219]. Ultimately, this demonstrates how *Drosophila* can also be an effective tool to screen antivirals for unwanted neurological side-effects, but there are multiple factors to consider when conducting drug studies in *Drosophila* (see Box 2).

Box 2The challenge of drug dosing in *Drosophila*.A recent study has detailed a standardized method for preparing food for drug administration in *Drosophila* and addressed several major challenges of drug dosing, including discrepancies in the literature about details of food and drug food preparations, impacts of food formulations on fly health, and potential issues in incorporating drugs into food stocks for the flies [220]. Many studies lack replicable detail regarding their food formulation by stating the use of “standard” *Drosophila* food/medium, which varied between studies, and some have varying formulations between food stocks and drug-infused food [221,222,223,224]. There are also discrepancies in the concentration of carrier solvent (such as DMSO) in fly food and its reported impact on or toxicity in *Drosophila* [221,225]. A compounding issue is that diet and the common use of “minimal/standard” media in drug preparations vastly impacts *Drosophila* in terms of physiology, morphology, lifespan, fecundity, and behaviour [226,227,228]. There are also many factors that could reduce the concentration and homogeneity of drugs being incorporated into fly food during preparation, which include heat, moisture, solvent use, storage procedures and length of time before use, reactivity of drugs with food components, among other factors, which could lead to drug decomposition before or during drug administration procedures with *Drosophila* [220,229]. Logistical issues must be considered as well, including assay size, when to administer drugs, drug concentrations, and routes of administration [4,230]. Maintaining an efficient number of flies per vial and transferring flies to new vials regularly during drug studies will reduce overcrowding and starvation. Meanwhile, physiologically effective drug concentrations range greatly between 0.01–100 mM in *Drosophila*, as standardized dose conversion studies between humans and animals do not include *Drosophila* [231,232]. Lastly, though oral administration is the easiest and most-used method, microinjection into organs or vapor administration of drugs can be used but are prone to complications and/or require special equipment. All these factors must be considered when conducting antiviral or other types of drug administration studies in *Drosophila*, which emphasizes the need for optimized standard diets, drug preparations, and fly handling and protocols for consistent results within and between research studies.

## 4. Endogenous Retroviruses and Transposons in *Drosophila*

Apart from transmissible viruses, *Drosophila* also carry endogenous viral elements in their genomes. Transposable elements (TEs) are grouped based on their mode of transmission. Class I TEs, such as retrotransposon (RTE) and endogenous retroviruses (ERVs), use a copy-and-paste mechanism to insert themselves into the host genome through reverse transcription of a viral genomic RNA intermediate into a DNA copy that is then inserted into a new genomic location. Class II TEs, such as DNA transposons, use a cut-and-paste mechanism, whereby their transposase enzyme extracts their DNA from the current genomic location and inserts it into a new position in the host genome. The readily quantifiable activity of fly transposable elements (TEs) allows researchers to study the role of these viral elements in fundamental cellular processes and disease models. Albeit less than within the human genome, it is estimated that 15% of the *Drosophila* genome houses TEs [233].

TE mobilization can be triggered by hybrid dysgenesis (due to asymmetric piRNA profiles), environmental changes, stress and proteinopathy (due to interference with siRNA silencing [234,235]. Moreover, exogenous viral infections in flies can modulate the expression of TEs through modulation of piRNA and siRNA profiles [236]. In wild-type flies, SINV infection caused a decrease in TE transcript load in somatic tissues, whereas infection of flies carrying loss-of-function mutations in the siRNA pathway resulted in enhanced TE levels. In contrast, germline tissue infected with Sigma virus (DMelSV), which has tropism for ovarian tissue, failed to drive a substantive change in TE expression [237]. This was postulated to be related to the more abundant TE-specific RNAi response in the ovary and immune pathways triggered upon infection. Indeed, transcript levels of the ERV *tirant* in *D. simulans* ovary are strongly correlated with relative piRNA amounts, further supporting that there is a strict control over TEs in the germline [238].

Another important aspect of ERVs is their ability to amplify both through cell-associated (retrotransposition) and cell-free (virion) mechanisms. Using S2 cells, it was shown that Gypsy [239] endogenous retrovirus can use both these modes of viral transmission; however, cell-to-cell infection is dependent on encoding a functional viral envelope protein, whereas intracellular retrotransposition events are independent of *env* [121]. This study suggests that the observed neuropathology of ERVs in neurodegenerative disease, such as Amyotrophic Lateral Sclerosis (ALS) [240,241] and dementia [242], may propagate throughout cerebral tissues through neuroinflammatory processes, driving cell-associated ERV expression as well as cell-free means of spread.

### 4.1. TEs through the Ages—From Development to Aging

Neuronal heterogeneity in the brain is in part due to the activity of TEs. Non-heritable de novo transposition in neural tissue increases TE load in individual cells [243,244] and is associated with neurological disease [245]. Additionally, the use of a combination of genome sequencing and single-cell mRNA sequencing in *Drosophila* midbrain led Treiber and Waddell to also propose that TEs introduce cryptic splice sites into nascent mRNA, which further broadens the neural transcriptome [246]. Thus, the genomic abundance and location of TEs could contribute to neural heterogeneity, inter-individual variation in neural function and neurological pathology.

Timing of TE expression can also impact neurophysiology. In mammalian systems, depletion of TRIM28 results in enhanced neuronal ERV levels [247,248]. During murine development, TRIM-28 depletion drove enhanced ERV expression in excitatory neurons and was associated with activated microglia and deposits of ERV-derived proteins [247]. Thus, early perturbation of ERV silencing may lead to an inflammatory response in the adult brain, and this has implications for neurodevelopmental and neurodegenerative disorders. However, there is also evidence for progressive loss of control over TEs during the typical aging process. In *Drosophila*, there is an age-dependent increase in LINE-like *R2* and RTE *gypsy* in brain [249,250]. Moreover, several studies show that disruption of the RNAi pathway (such as piwi-null or Ago2-null mutants) can enhance TE expression during aging [249,250], in addition to increasing the genomic TE load [250]. Conversely, knock down of PAF1 (polymerase-associated factor; antagonizes RNAi pathways) improved suppression of TEs during aging in flies [250]. In general, control over TE expression during aging also impacts fly lifespan, with a higher load of TE activity associated with decreased longevity [249,250], except for telomeric TEs whose activity is more abundant in long-lived *Drosophila melanogaster* strains [251]. Again, genomic abundance and location of TEs may play an important role in diseases of aging and overall longevity.

### 4.2. ERVs, TDP-43 Pathology and Motor Neuron Disease

Flies have been widely employed to understand ALS-risk genes [252], which ties to the fact that select ALS subtypes are associated with elevation of RTEs and ERVs [253]. An examination of *Drosophila* ERVs has also highlighted their putative role in neurodegenerative disease processes. Physiologically, ALS-associated risk gene TDP-43 (TBPH in flies) plays a role in limiting RTE activity. Robust enhancement of RTE expression is observed in TBPH-null flies, whereas rescue by genetic expression of the TBPH protein repressed RTE expression in the TBPH-null background [254]. Mechanistically, disruption of TBPH led to defects in the siRNA pathway through regulating the expression of Dicer-2 levels, resulting in upregulation of Gypsy and motor neuron degeneration [254]. Moreover, oral dosing of flies with a panel of reverse transcriptase inhibitors reversed locomotor deficiencies in TBPH-null 3rd instar larvae. Josh Dubnau’s group has also been at the forefront of pathological examination of Gypsy in flies and its role in neurodegeneration. Using a Drosophila model of human TDP-43 expression in neurons and glia, it was shown that TDP-43 proteinopathy causes an increase in RTEs, including Gypsy expression [234,255]. TDP-43 proteinopathy in flies resulted in locomotor problems and premature death. An underlying driver of RTE-mediated cell death was DNA damage, possibly due to attempted RTE re-integration events [234,255]. The use of a Gypsy-CLEVR reporter system further allowed quantification of RTE-mediated DNA damage through labeling histone γH2AV (fly equivalent of human γH2AX) foci indicative of double-stranded DNA breaks in cells with Gypsy activity [255,256]. Further demonstrated in this hTDP-43 model was that glial apoptosis driven by RTE activity was protective for nearby neurons [255]. This effect on neuronal survival is directly related to an increase in Gypsy production upon inhibition of apoptosis in hTDP-43^+^ glia, thus resulting in enhanced RTE-mediated DNA damage in adjacent neurons [255]. In a follow-up study, SF2-null flies (homologue of human SRSF1; an RNA splicing factor) crossed with *RepoTS>TDP43* strain displayed improved locomotion in negative geotaxis assays and enhanced lifespan, indicating a rescue of TDP-43 pathology in glial cells [257]. Considering SRSF1’s known roles in HIV transcription [258] and as a mediator in C9orf72 ALS [259], this work speaks to the role of TDP-43 pathology in the deregulation of ERVs and motor neuron disease, and the involvement of potentially many other co-factors in modulating the expression and control of RTEs.

### 4.3. TEs and Tauopathy

Tau pathology is found in several neurodegenerative conditions and is associated with neurocognitive decline and/or motor impairment. Tauopathy is caused by the misfolding and aggregation of Tau protein, which leads to a loss of function and proteinopathy. Inducing expression of a mutant form of human Tau by crossing transgenic tau^R406W^
*Drosophila* with a pan-neuronal driver results in progressive neurodegeneration in the progeny [260]. Similarly, neuronal expression of Tau^WT^ or Tau^E14^ flies exhibit Tau neurotoxicity [261,262]. In all cases, these Tau transgenic flies exhibit evidence of global nuclear chromatin relaxation [263], which has been associated with enhanced TE activity. Indeed, both Tau^WT^ and Tau^R406W^ flies have elevated expression of TEs, including *copia*, *gypsy* and L1-like non-LTR RTEs (*Het-A*) [264]. These results mirror observations of enhanced TE levels in the dorsolateral prefrontal cortex of patients with neuropathologic diagnosis of Alzheimer’s disease (AD) as compared to controls [264]. Moreover, select ERV expression was associated with poorer global cognitive performance in patients in the year preceding death [264], suggesting ERV burden is linked with the severity of dementia.

### 4.4. TEs and Huntington Protein Expansion

Likewise, a fly model of Huntington’s disease (HD), whereby a pathogenic variant of human Huntington protein (128QHttFL) is expressed pan-neuronally, also exhibits derepression of TEs [265], but not in flies expressing wild-type nonexpanded Huntington 16QHtt. Early TE transcript load was correlated with age-dependent neurodegeneration, as measured by loss of the neuronal marker Elav, indicating neuronal loss. *Repo-Gal4>128QHtt* flies showed a distinct pattern of TE expression from that of *Elav-Gal4>128QHtt* flies, in conjunction with severe locomotor deficit and decreased survival. The molecular mechanism behind mutant Huntington activation of TEs was loss of histone H3 methylation (H3K9me3), leading to global heterochromatin relaxation. Flies fed with reverse transcriptase inhibitors were rescued from altered HD eye phenotype in all 128QHttFL transgenics, indicating that TE activity was driving neuropathology, and more specifically genomic DNA damage as measured by accumulation of γH2Av-positive foci in larval brains. RNA-seq analysis of human HD tissues corroborates an elevation in TEs [266]. Together, this indicates that either neuronal or glial expression of mutant Huntington drives TE-mediated neuropathology, and further supports the notion that *Drosophila* are a useful model system for examination of neurological disease.

## 5. Conclusions

While flies have been widely adopted for studying neurological development, processes and disorders [75,267,268,269], they have been underutilized as a model for neurological infection. When speed and cost are at a premium, *Drosophila* models fulfill this need through their ease of genetic manipulation and animal husbandry. As described above, the variety of possible approaches to address viral infection and immunity in neurological disease allows for a gateway into addressing questions about known and emerging pathogens, as well as genome-encoded viruses. Additionally, *Drosophila* allows for examining exquisite differences in viral strain or viral proteins in the context of a variety of genetic backgrounds, while also assessing environmental variables and a spectrum of developmental stages given the fly’s short lifespan. Moreover, the standardized and reproducible readouts of fly behaviour, motor function or neurodegeneration permit an accurate assessment of neurological outcomes.

These benefits of neurological infection modelling in flies must also be balanced with the limitation that their neurological and immunological systems are not as complex as mammals. Depending on the virus studied, certain aspects of pathophysiology in humans cannot be recapitulated in flies, and studies should highlight these caveats. Moving forward, considering *Drosophila* as a valuable model system for neurological infections may guide the discovery of many novel virus–host interactions.

## Figures and Tables

**Figure 1 cells-11-02685-f001:**
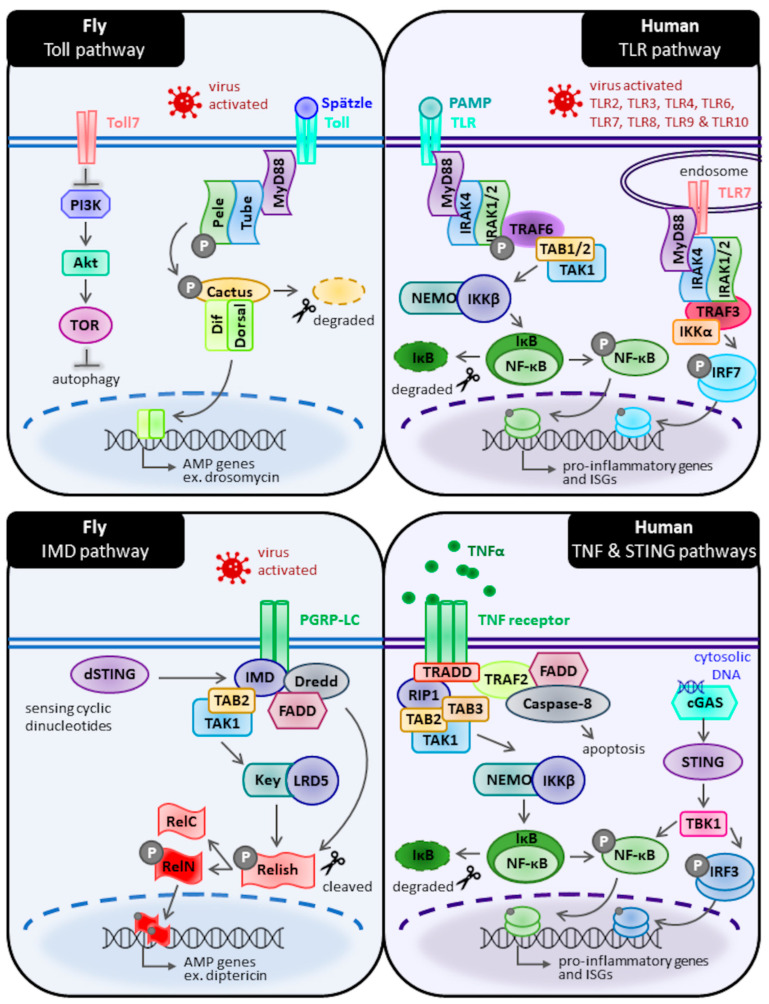
Comparison of *Drosophila* and human Toll/TLR and IMD/TNF signalling cascades. Basic diagram of Toll signalling in *Drosophila* versus Toll-like receptor (TLR) signalling in humans (top). Likewise, simplified diagram of IMD signalling in *Drosophila* versus TNFα receptor and cGAS/STING signalling in humans (bottom). Homologies between fly and human signalling proteins are shown through similar colour and shape. A detailed description of each pathway and its key proteins is included in the main text. Note that not all known signalling proteins and adaptors are included in the diagram for the sake of simplicity.

**Figure 2 cells-11-02685-f002:**
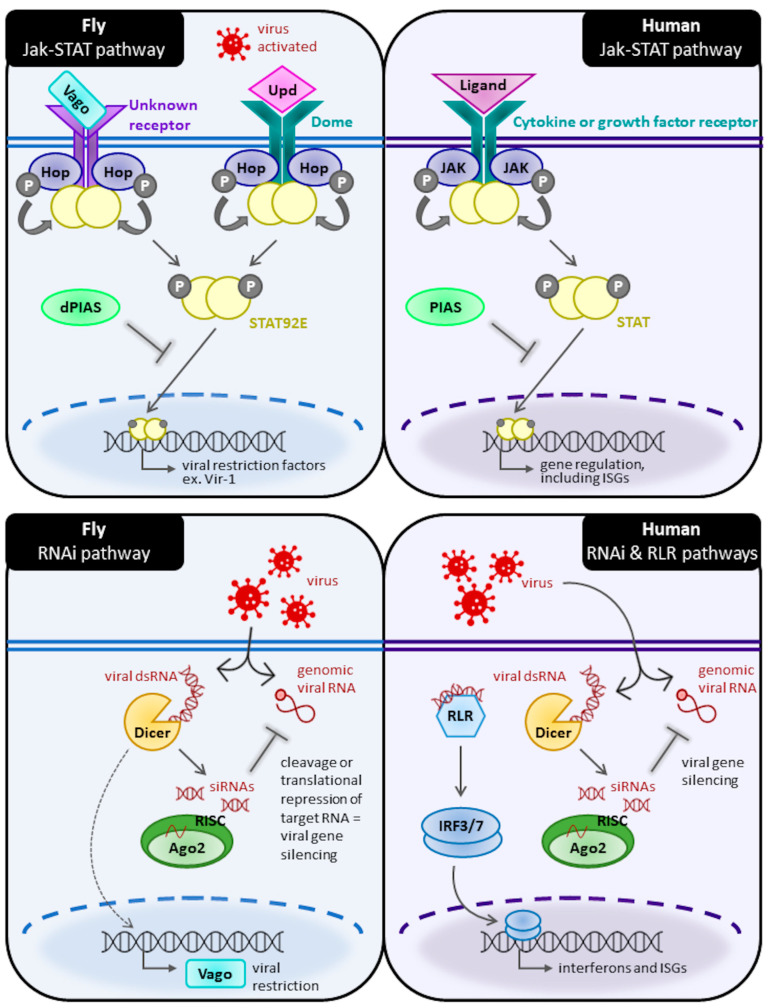
Comparison of *Drosophila* and human Jak/STAT and RNAi signalling cascades. Basic diagram of Jak-STAT signalling in *Drosophila* versus humans (top). Likewise, simplified diagrams of RNAi signalling in *Drosophila* versus RNAi and RIG-like receptor (RLR) signalling in humans (bottom). Homologies between fly and human signalling proteins are shown through similar colour and shape. A detailed description of each pathway and its key proteins is included in the main text. Note that not all known signalling proteins and adaptors are included in the diagram for the sake of simplicity.

**Figure 3 cells-11-02685-f003:**
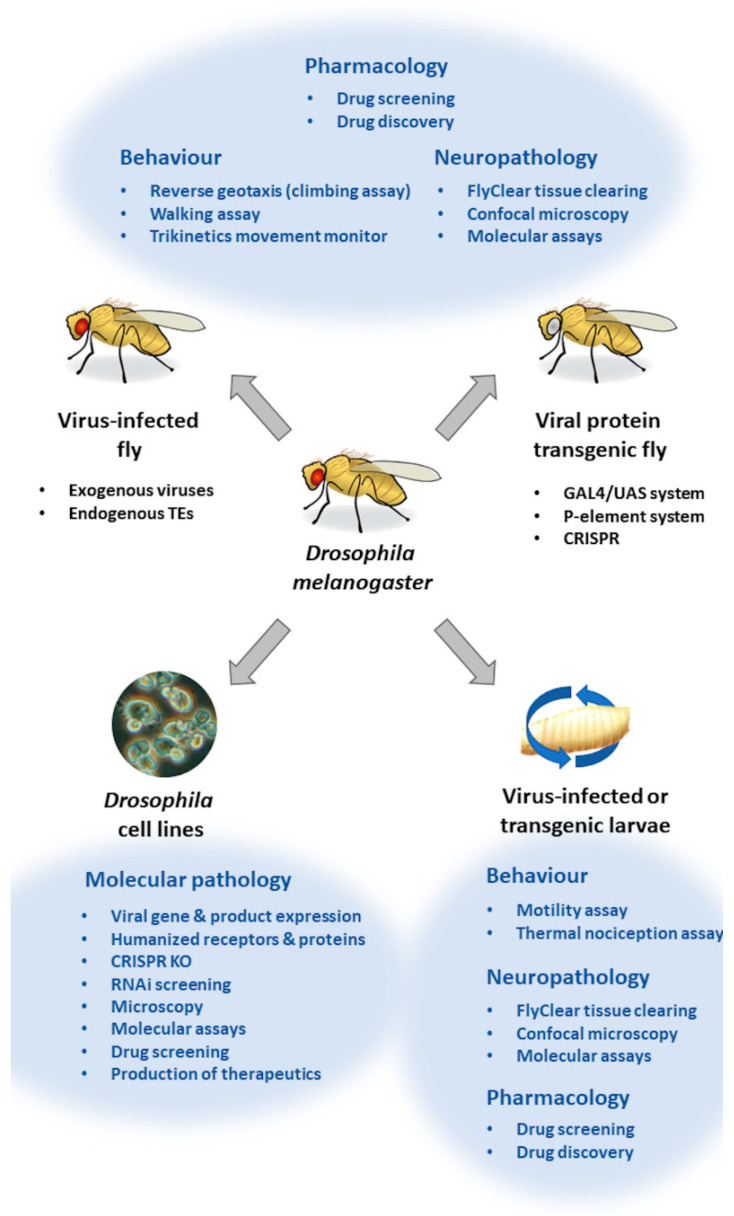
Strategies to study neuroinfections using *Drosophila* models. Virus–host interactions leading to neurological sequelae can be studied in flies using *Drosophila* cell lines, larvae or flies infected with exogenous or endogenous viruses and through viral transgenic systems. Outcomes of in vivo infection in larvae and adult flies can be examined through a variety of behavioural, neuropathological and pharmacological approaches. *Drosophila*-derived cell lines can further be used to delineate molecular processes associated with viral infection and host immunity, due to the ease of expression of viral or cellular proteins, or through genetic or cellular manipulations using CRISPR and RNAi, respectively. Examples of these approaches can be found throughout the main text.

**Table 1 cells-11-02685-t001:** *Drosophila* antiviral and immunity genes with homology in humans.

Fly Pathway	*Drosophila* Gene	Human Pathway	Human Gene; Protein Name	FlyBase ^a^	Literature ^b^	Reference
Toll	Cactus	TLR	NFKBIA; IκBα	O	H	[17,18]
	Dif		NFKB1; NF-κB p50/RELA; NF-κB p65		H	[12,18,19]
	Dif		RELB	O		
	Dorsal		NFKB1; NF-κB p50		H	[20]
	Dorsal		RELA; NF-κB p65	O		
	MyD88		MYD88	O	O	[21,22]
	Pelle		IRAK1		O	[21,23]
	Pelle		IRAK4	O		
	Pellino		PELI1 & PELI2	O	H	[24,25]
	Tube		IRAK4		O	[21,23]
	Tube		MAL		H	[22]
IMD	Bendless	TNFR	UBE2N; UBC13	O	H	[26]
	Dredd		CASP8; Caspase-8		H	[26,27,28,29]
	Dredd		CASP10	O		
	Fadd		FADD	O	O	[26,30]
	Iap2		BIRC3; cIAP2		H	[29,31]
	Iap2		BIRC2; cIAP1	O		
	Imd		RIPK1; RIP1		H	[27,32,33]
	Ird5		IKBKB; IKKβ	O	H	[34,35,36,37]
	Ird5		CHUK; IKKα	O		
	Kenny; IKKγ		IKBKG; NEMO/IKKγ	O	H	[34,38]
	Kenny; IKKγ		OPTN	O		
	Relish		NFKBIA; IκBα/ NFKB1; NF-κB p50/RELA; NF-κB p65	O	H	[39,40]
	Tab2		TAB2 & TAB3	O	H	[11,31]
	Tak1		MAP3K7; TAK1	O	H	[31,41]
	UEV1a		UBE2V1; UEV1a		H	[26]
	UEV1a		UBE2V2; UEV2	O		
	dSTING	cGAS/STING	STING1; STING	O	O	[42]
Jak-STAT	Cg14225/Latran	Jak-STAT	IL6ST; GP130		H	[43,44]
	Domeless		LIFR & CNTFR		H	[43,45]
	Domeless		PTPRQ	O		
	Hopscotch		JAK1 & JAK2	O	H	[46]
	Marelle		STAT5A/STAT5B; STAT5 & STAT6	O	H	[47]
	Socs36E		SOCS4 & SOCS5	O	H	[48,49]
	STAT92E		STAT3 & STAT5A/STAT5B; STAT5	O	H	[47,50]
	Su(Var)2-10/dPIAS		PIAS1	O	H	[51,52]
RNAi	Ago-1, Ago-2, & Ago-3	RNAi	AGO1, AGO2, & AGO3		H	[53,54]
	Ago-1		AGO2	O		
	Ago-3		PIWIL2	O		
	Armitage		MOV10L1	O	H	[55]
	Aubergine		PIWIL1; Hiwi	O	H	[53]
	Dicer		DICER1	O	H	[56]
	Fmr1		FMR1	O	H	[57,58]
	Piwi		PIWIL1; Hiwi		H	[59]
	Piwi		PIWIL3; Hiwi3	O		
	R2d2		TARBP2		H	[60]
	Rm62/Dmp68		DDX5; P68		H	[58]
	Vasa intronic gene		SERBP1; PAI-RBP1	O	H	[61,62]
Restriction factors	Dcp2	Restriction factor	DCP2	O	H	[63]
	Ge-1		EDC4; RCD-8	O	H	[64]
	FoxK		FOXK1 & FOXK2	O	O	[65]
	Nazo		C19orf12		O	[66]
	Nup98		NUP98	O	H	[65,67,68,69]
	Ref(2)p		SQSTM1; P62	O	H	[70,71]
	TREX		TREX1	O	H	[72,73]

^a^ FlyBase human orthologs (O) listings with genes denoted as “Yes” for both Best Score and Best Reverse Score. ^b^ Cited in literature as homolog (H) or ortholog (O).

## Data Availability

*Drosophila* gene names and ontology can be found in the FlyBase database at: www.flybase.org. Human gene names can be found at: www.genecards.org.

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
