# Peer review of "Drosophila as a Model for Human Viral Neuroinfections"

_cells, 2022, doi:10.3390/cells11172685_

Round 1

Reviewer 1 Report

Due to the genetic and physiological similarities, majorities of virus studies were done in rodents and non-human primates. Accumulating studies have shown that Drosophila could be a good tool in virus infection and pathogenesis studies. In this review article, Benoit and colleagues summarized the virus infection models in Drosophila. The authors summarized the advantage and limitations of the Drosophila model. This review is comprehensive, well-written, and up-to-date. I only have minor concerns:

1.       Some references are missing in Table 1.

2.       Most of the references about in vitro studies used S2 cells, which are macrophage-like cells, and the in vivo studies are not really focused on the brain. I would suggest changing the titer to “Drosophila as a Model for Human Viral Infections”.

3.       Check typos in the manuscript.

Author Response

Response to all reviewers and editors Dr. Brandenburg and Dr. Bei Zhang:

We thank the reviewers for their thoughtful comments for improvement of our manuscript.  We have performed the suggested modifications, addressed reviewers’ comments, and corrected the text and figures accordingly.

Specific Response to Reviewer #1:

Due to the genetic and physiological similarities, majorities of virus studies were done in rodents and non-human primates.  Accumulating studies have shown that Drosophila could be a good tool in virus infection and pathogenesis studies. In this review article, Benoit and colleagues summarized the virus infection models in Drosophila. The authors summarized the advantage and limitations of the Drosophila model. This review is comprehensive, well-written, and up-to-date. I only have minor concerns.

 Comments and point-by-point responses:

1) Some references are missing in Table 1.

 1)  We would like to provide a clarification regarding referencing in Table 1.  References in Table 1 reflect protein homology annotations from the literature, as indicated by footnote “b”.  When protein homology was indicated by FlyBase only (footnote “a”), no reference was assigned to that row.   

2) Most of the references about in vitro studies used S2 cells, which are macrophage-like cells, and the in vivo studies are not really focused on the brain.                 

2)  As immune cell infiltration into the CNS is an important aspect of neuropathology, the in vitro studies using S2 macrophage-like cells are valuable to improving our understanding of neurological viral infections, despite not being a resident brain cell.  To better indicate this to our readers, we have added text on lines 272-274 stating, “they are essential for control of virus infection and brain immunity”.

3)  I would suggest changing the titer to “Drosophila as a Model for Human Viral Infections”.

 3)  The authors respectfully disagree and feel that the current title indicates to the reader that the review focuses on neurological infections. We have left the title as previously worded.

 4)  Check typos in the manuscript. 

 4)  Thank you for this suggestion.  We have proof-read the article and edited as indicated by track changes. 

Reviewer 2 Report

The manuscript by Benoit et al. represents an excellent overview over past and present applications of Drosophila melanogaster in studying and modelling viral infections of the nervous system. The authors have done an excellent job in covering the field in this well-structured, well written and organised work. The only criticism, I would suggest the authors to consider in a revision of their manuscript, is to make their article more accessible for non-Drosophila readership.

To improve the accessibility for the non-fly readership the authors could

- present some schematic cartoons of the major immunity response pathways in comparison to human pathways as an addition to section 2 (line 43 and following). At present the authors only present a table that compares the fly and human homologs. I think that a table is a good reference to indicate conservation of molecular players, however is less instructive in comparing the overall pathway similarity. This would not have to be done to every detail, but the core pathways should be compared between flies and humans (as the text already indicates).

- in particular the direct comparison of the IMD pathway would be useful (section 2.2.) to be shown as a schematic. This would help the reader to orient themselves in the manuscript, when reading the more specific sections on viral response pathways and their interactions.

- The Gal4/UAS system is explained (line 568) only after it was first mentioned in a context; therefore in lines 462, 463 is remains obscure what these Gal4 drivers are and how the methods works. 

other comments:

in line 299 following, the authors mention that the lack of certain viral response pathways (e.g. entry or replication) can be a caveat in using flies as a model. In a reverse argument, this could be of an advantage when examining the requirements of certain molecular components in these responses.

in line 587 following, the authors describe the use of the fly compound eye as a genetic test tube to examine the interaction between EBV immediate early genes and and host genes. It would be nice to have this illustrated with a figure showing normal and rough eye phenotypes to demonstrate the beauty and ease of genetic analyses on adult phenotypes in the fly.

Author Response

Response to all reviewers and editors Dr. Brandenburg and Dr. Bei Zhang:

We thank the reviewers for their thoughtful comments for improvement of our manuscript.  We have performed the suggested modifications, addressed reviewers’ comments, and corrected the text and figures accordingly.

Specific Response to Reviewer #2:

The manuscript by Benoit et al. represents an excellent overview over past and present applications of Drosophila melanogaster in studying and modelling viral infections of the nervous system. The authors have done an excellent job in covering the field in this wellstructured, well written and organised work. The only criticism, would suggest the authors to consider in a revision of their manuscript, is to make their article more accessible for nonDrosophila readership. To improve the accessibility for the non-fly readership the authors could:

  Comments and point-by-point responses:

 1) - present some schematic cartoons of the major immunity response pathways in comparison to human pathways as an addition to section 2 (line 43 and following). At present the authors only present a table that compares the fly and human homologs. I think that a table is a good reference to indicate conservation of molecular players, however is less instructive in comparing the overall pathway similarity. This would not have to be done to every detail, but the core pathways should be compared between flies and humans (as the text already indicates).

1)  Thank you for this suggestion by reviewers #2 and #3.  We have now added new Figure 1 (lines 104-112) and Figure 2 (lines 213-220) to facilitate comparison between fly and human immunity pathways.  Figure 1 shows a comparison of Drosophila and human Toll/TLR and IMD/TNF signalling cascades, whereas Figure 2 shows a comparison of Drosophila and human Jak/STAT and RNAi signalling cascades.  We hope this addition to our manuscript assists non-Drosophila experts appreciate the signalling pathways discussed in the main text.

 2) - in particular the direct comparison of the IMD pathway would be useful (section 2.2.) to be shown as a schematic. This would help the reader to orient themselves in the manuscript, when reading the more specific sections on viral response pathways and their interactions.

 2)  Thank you for this suggestion.  We have now added new Figure 1 (bottom) depicting a comparison of IMD/TNF signalling cascades, with an addition of how STING signalling converges into the IMD pathway in flies, but is independent of TNF signalling in humans.  We hope this addition to our manuscript assists non-Drosophila experts appreciate the signalling pathways discussed in the main text.

3)  - The Gal4/UAS system is explained (line 568) only after it was first mentioned in a context; therefore in lines 462, 463 is remains obscure what these Gal4 drivers are and how the methods works.

3)  Thank you for pointing this out.  We have added text on lines 492-496 to better explain the Gal4/UAS system prior to its first mention in the text (former line 462, new line 496).

4)  in line 299 following, the authors mention that the lack of certain viral response pathways (e.g. entry or replication) can be a caveat in using flies as a model. In a reverse argument, this could be of an advantage when examining the requirements of certain molecular components in these responses.

 4)  Thank you for mentioning this.  We have edited the text on lines 327-328 to better indicate that these biological differences can be both a caveat and benefit as well, depending on the experimental design.   

5)  in line 587 following, the authors describe the use of the fly compound eye as a genetic test tube to examine the interaction between EBV immediate early genes and and host genes. It would be nice to have this illustrated with a figure showing normal and rough eye phenotypes to demonstrate the beauty and ease of genetic analyses on adult phenotypes in the fly.

 5)  We appreciate the need for the readership to have a visualization of the rough eye phenotype and therefore have referred them to an excellent example in reference #212.  Please see edits on lines 670-671.

Reviewer 3 Report

Review article entitled “Drosophila as a Model for Human Viral Neuroinfections” by Benoit et al. Focusing on benefits of Drosophila neural system for neurological infections and neuroimmunity. Overall rereview article is excellent. Authors summarised the antiviral and immunity genes of Drosophila with homology to humans and discussed the overview of antiviral immune signalling pathways. Authors Highlighted different in vivo and in vitro studies using Drosophila for the virus associated with neurological diseases. However, I have some suggestions to improve the quality of the review article.

1.     Authors discussed the antiviral immune signalling pathway in text but if authors also make figures for all four pathways individually or a single figure for all four pathways make more clarity.

2.     Authors can make a table for the endpoint of viral Neuroinfections in human and Drosophila model

3.     Authors discussed in detail the benefit of use of the Drosophila models for the viral neuroinfections studies but not the limitation of the model organism. Please discuss in detail.  

Author Response

Response to all reviewers and editors Dr. Brandenburg and Dr. Bei Zhang:

We thank the reviewers for their thoughtful comments for improvement of our manuscript.  We have performed the suggested modifications, addressed reviewers’ comments, and corrected the text and figures accordingly.

Specific Response to Reviewer #3:

Review article entitled “Drosophila as a Model for Human Viral Neuroinfections” by Benoit et al. Focusing on benefits of Drosophila neural system for neurological infections and neuroimmunity. Overall rereview article is excellent. Authors summarised the antiviral and immunity genes of Drosophila with homology to humans and discussed the overview of antiviral immune signalling pathways. Authors Highlighted different in vivo and in vitro studies using Drosophila for the virus associated with neurological diseases. However, I have some suggestions to improve the quality of the review article.

 Comments and point-by-point responses:

1) Authors discussed the antiviral immune signalling pathway in text but if authors also make figures for all four pathways individually or a single figure for all four pathways make more clarity.

 1)  Thank you for this suggestion by reviewers #2 and #3.  We have now added new Figure 1 (lines 104-112) and Figure 2 (lines 213-220) to facilitate comparison between fly and human immunity pathways.  Figure 1 shows a comparison of Drosophila and human Toll/TLR and IMD/TNF signalling cascades, whereas Figure 2 shows a comparison of Drosophila and human Jak/STAT and RNAi signalling cascades.  We hope this addition to our manuscript assists non-Drosophila experts appreciate the signalling pathways discussed in the main text. 

2) Authors can make a table for the endpoint of viral Neuroinfections in human and Drosophila model.

2)  As an excellent table similar to this already exist in the literature, we have now highlighted and referred the readership to an article by Xu and Cherry (reference #16) on lines 444-446 of the main text.

3)  Authors discussed in detail the benefit of use of the Drosophila models for the viral neuroinfections studies but not the limitation of the model organism. Please discuss in detail.

 3)  Thank you for mentioning this.  We have edited the text on lines 42-48 to better indicate the limitations of Drosophila as a model species.